# Phytochemicals-linked food safety and human health protective benefits of the selected food-based botanicals

**Ashish Christopher, Kalidas Shetty****\***

Department of Plant Sciences, North Dakota State University, Fargo, North Dakota, United States of America

\* kalidas.shetty@ndsu.edu

## Abstract

Phytochemicals-rich food-based botanicals including traditional or under-utilized plant-based ingredients can serve a dual functional role to help counter food contamination of bacterial origin, while also addressing the rise of diet-linked non-communicable chronic diseases (NCDs) such as type 2 diabetes, chronic hypertension and the associated oxidative stress. Hence the screening of these food-based botanicals for their phenolic content and profile, as well as antimicrobial, antioxidant, anti-hyperglycemic and anti-hypertensive properties has relevant merit. Using *in vitro* assay models, hot water extracts of different forms (slice, pickle, or powder) of amla (*Phyllanthus emblica*), clove (*Syzygium aromaticum*), kokum (*Garcinia indica*), and garlic (*Allium sativum*) were analyzed for their total soluble phenolic content (TSP) and phenolic profile as well as antimicrobial activity against strains of *Salmonella Enteritidis*, *Listeria monocytogenes*, and *Escherichia coli* that are associated with food-borne disease outbreaks. In addition, the antioxidant, anti-hyperglycemic and anti-hypertensive activity of the extracts were also determined using *in vitro* assay models, with the goal of establishing a dual functional role of the food safety and health protective benefits of these botanicals. A high baseline TSP content was observed in all the extracts and the major phenolic phytochemicals detected were gallic, cinnamic, ellagic, benzoic, dihydroxy-benzoic, protocatechuic, and *p*-coumaric acid along with catechin and rutin. All extracts displayed significant antimicrobial activity against most of the bacterial strains tested and the antimicrobial activity was specific for each strain targeted in this study. Furthermore, significant antioxidant, anti-hyperglycemic and antihypertensive activity were observed among the botanical extracts, especially among the amla and kokum extracts. These results indicate that phytochemicals enriched botanicals, including amla and kokum, can be integrated into modern-day food preservation and dietary support strategies aimed at improving the food safety and health protective benefits of the food matrix.

**Data Availability Statement:** All relevant data are within the manuscript and its Supporting Information files.

**Funding:** The author(s) received no specific funding for this work.

## Introduction

Food-based botanicals including fruits, vegetables and spices often have a high content of plant secondary metabolites or phytochemicals that range from simple phenolic acids to

**Competing interests:** The authors have declared that no competing interests exist.

complex polyphenolic compounds. The rich phenolic phytochemicals profile and associated bioactivity of these food-based botanicals makes them suitable for incorporation into dietary strategies that are aimed at addressing important public health challenges, such as the persistence of foodborne diseases caused by bacterial pathogens, and the rise in NCDs such as type 2 diabetes, chronic hypertension and the oxidative stress that is usually associated with these NCDs. In this regard the screening of such food-based botanicals, including underutilized or traditional foods, still has relevant merit as evident by current literature on the bioactivity of such food-based botanicals [1,2].

Amla (*Emblica officinalis* Gaertn or *Phyllanthus emblica* Linn), commonly known as Indian gooseberry, is an important traditional fruit and medicinal plant in the Indian Subcontinent and other parts of the South-East Asia. This underutilized fruit is widely found in India, Pakistan, Sri Lanka, China, Uzbekistan, and Malaysia [3]. Amla is known to possess a wide range of bioactive properties including antioxidant, antidiabetic, and antimicrobial activity [4–10]. *In vitro* studies have shown amla to possess antimicrobial activity against gram-positive (*Staphylococcus*, *Micrococcus*, and *Bacillus*) and gram-negative (*E. coli* and *Salmonella*) bacterial pathogens [11–13]. Due to its high bioactivity, amla has the potential to be utilized as a dual functional food ingredient in different food formulations for the improvement of food safety and human health benefits. Previous research has targeted the incorporation of amla into mixed fruit beverages, pan bread, and chicken feed to improve functional properties of the food matrix [14–18].

Kokum (*Garcinia indica*) or mangosteen is an indigenous fruit commonly found in India and has many phytopharmacological properties with wider food, pharmaceutical, and industrial applications [19–23]. The major phytochemicals present in kokum that are responsible for their wider bioactivity include garcinol, hydroxycitric acid (HCA), cyanidin-3-sambubioside, and cyanidin-3-glucoside [20–22,24]. Extracts of kokum were found to have antimicrobial activity against *Micrococcus aureus*, *Bacillus megaterium*, *Micrococcus luteus*, *Salmonella Typhimurium*, *Pseudomonas aeruginosa*, *E. coli*, *Bacillus subtilis*, *Enterobacter aerogenes*, and *Staphylococcus aureus* [25,26].

Clove (*Syzygium aromaticum*), a native of the Maluku Islands in Indonesia, is one of the most valuable spices that has been used for centuries as a food preservative and for other medicinal purposes. This spice is currently grown in several countries including Indonesia, India, Malaysia, Sri Lanka, Madagascar, Tanzania, and Brazil [27]. Clove has a high antioxidant activity which is mostly due to its flavonoid (e.g., quercetin) and essential oil (e.g., eugenol) content [28–30]. Clove also has potential antihyperglycemic properties through the inhibition of carbohydrate digestive enzymes such as α-amylase and α-glucosidase, which is a common therapeutic target to manage chronic hyperglycemia [31,32]. Clove (flower buds) and clove essential oils (e.g., eugenol) have shown antimicrobial activity against different gram-positive (*Staphylococcus*, *Listeria*, and *Bacillus*) and gram-negative (*Enterobacter*, *Shigella*, *E. coli*, *Salmonella*, *Klebsiella*, and *Pseudomonas*) bacterial pathogens, thereby making them valuable natural preservatives with potential for different food safety applications [33–37].

Similarly, garlic (*Allium sativum*) is another example of a botanical with phytopharmacological properties that include antioxidant, antihypertension, antidiabetic, and antimicrobial activity [38–43]. The bioactivity of garlic is mostly due to the presence of organosulfur compounds such as allicin and S-allylcysteine which are present at a concentration of around 2.3% [44–46].

Based on these wider antimicrobial and health-protective functional properties of the above botanicals, the goal of this study was to screen different forms of these selected food-aligned botanicals (powder, slice, or pickle) for their antimicrobial activity against strains of bacterial

pathogens (*E. coli*, *Listeria*, and *Salmonella*) that are associated with foodborne illnesses. In addition, the total soluble phenolic content, phenolic profile, antioxidant activity, and enzyme inhibitory activity against type 2 diabetes-relevant α-amylase and α-glucosidase and hypertensive-relevant angiotensin-I-converting enzyme (ACE), were also investigated. The results of this study will help determine the food safety-relevant and human health protective functional benefits of these food-based botanicals which can be utilized in future dietary strategies to help counter foodborne illnesses due to microbial contamination, as well as to help mitigate and manage the prevalence of type 2 diabetes and chronic hypertension associated with diet-linked NCDs.

## Materials and methods

### Chemicals used

Other than mentioned, all chemicals and enzymes were purchased from Sigma Chemical Co (St. Louis, MO, USA).The chemicals used in this study were Folin-Ciocalteu reagent, 95% ethanol, absolute methanol, sodium carbonate, gallic acid, benzoic acid, protocatechuic acid, ellagic acid, cinnamic acid, dihydroxybenzoic acid, p-coumaric acid, rutin, catechin, phosphate buffer saline, 2, 2-Dipheny-1-Picryl Hydrazyl (DPPH), 2, 2-Azino-bis-(3-ethylbenzthiazoline-6-sulfonic acid) (ABTS), sodium phosphate buffer, potassium phosphate buffer, brain heart infusion broth, Luria-Bertani (LB) broth, Mueller-Hinton (MH) broth, glycerol, sodium chloride, dinitro salicylic acid, acarbose, hippuryl-histidyl-leucine (HHL), sodium chloride-borate buffer, and hydrochloric acid. The enzymes used in this study were α-amylase, α-glucosidase and angiotensin-I-converting enzyme (ACE).

### Samples used

Clove (flower buds), amla (powder, slice, and pickle), kokum (dried slices), and garlic (slice and pickle) were purchased from a local Indian grocery store in Fargo (North Dakota, USA). The clove and kokum samples were ground using a coffee blender to obtain a coarse powder while the amla, kokum, and garlic samples (slice and pickle) were chopped into smaller pieces before extraction.

### Preparation of extracts

The extraction of all samples was done using hot water. For the clove, amla, and kokum samples (powder and pickle), 10 g of the sample was used in the hot water extraction protocol, while for the rest of the samples- amla, kokum, and garlic (slices), 25 g of the sample was used. For the hot water extraction protocol, 10 g or 25 g of the respective samples were added to 50 mL of boiling water (100˚C) and boiled for 15 min after which the samples were cooled down to room temperature and centrifuged at 8,500 rpm for 15 min. The supernatant was collected and re-centrifuged at 8,500 rpm for 15 min and the extracts were stored at -20˚C. For the antimicrobial assay, the frozen extracts were thawed at room temperature and filter-sterilized using 0.22 μm syringe filters (Millipore Corp, MA, USA) prior to the assay. The amla slice, amla pickle, garlic slice and garlic pickle extracts were analyzed on a fresh weight (FW) basis while the clove powder, amla powder, kokum slice and kokum powder extracts were analyzed on a dry weight (DW) basis.

### Bacterial strains used

The *Salmonella* strains tested in this study were *Salmonella enterica* subsp. *enterica* serovar Enteritidis (ATCC BAA-1045), *S. enterica* subsp. *enterica* serovar Typhimurium (FSL R8-

0865), *S. enterica* subsp. *enterica* serovar Montevideo (FSL R8-3417), *S. enterica* subsp. *enterica* serovar Stanley (FSL R8-3511), and *S. enterica* subsp. *enterica* serovar Saintpaul (FSL R8-3582). The *Listeria* strains analyzed in this study were *Listeria monocytogenes* serovar 1/2a (10403S), *L. monocytogenes* serovar 1/2b (FSL J1-0194), *L. monocytogenes* serovar 1/2a (FSL F2-0515), and *L. monocytogenes* serovar 4b (H7858). The *E. coli* strains used in this study were *E. coli* serovar O157:H7 (EDL-933) and *E. coli* serovar O26:H11 (TW07936). The *Salmonella* and *Listeria* stocks were stored at -80˚C in brain heart infusion broth (Oxoid, Basingstoke, UK) containing 25% glycerol. The *E. coli* stocks were stored at -80˚C in LB broth, Miller (Luria-Bertani) (Difco, Becton, Dickinson and Company, Sparks, MD, USA) containing 25% glycerol. The *Salmonella* and *Listeria* strains were obtained from the Food Safety Laboratory at Cornell University, USA, except for *Salmonella enterica* subsp. *enterica* serovar Enteritidis (ATCC BAA-1045) which was obtained from the American Type Culture Collection (Manassas, VA, USA). The *E. coli* strains were obtained from the Thomas S. Whittam STEC Center at Michigan State University, MI, USA.

## Total soluble phenolic (TSP) content

The TSP content of the extracts was determined using the Folin-Ciocalteu method based on a protocol as described previously [47]. For this assay, the extracts were diluted in water at 1:20 dilution and 0.5 mL aliquots of the diluted extracts were taken into respective glass tubes after which 1 mL of 95% ethanol, 0.5 mL of 50% (v/v) Folin-Ciocalteu reagent, and 1 mL of 5% sodium carbonate were added sequentially to each tube. Due to the higher ascorbic acid content of the amla, which interferes with the Folin-Ciocalteu reagent, we have used multiple dilutions of the botanical extracts to avoid overestimation of total soluble phenolic content. The tubes were then mixed using a vortex machine and incubated for 60 min under dark conditions. The absorbance values in each tube were measured at 725 nm with a UV-visible spectrophotometer (Genesys 10S UV-VIS spectrophotometer, Thermo Scientific, NY, USA). Using a standard curve of different concentrations of gallic acid (0.025 to 0.3 mg/mL) in 95% ethanol, the absorbance values of the extracts were converted, and the TSP content was expressed in milligram gallic acid equivalents per gram fresh weight or dry weight (mg GAE/g FW or DW). The amla slice, amla pickle, garlic slice and garlic pickle extracts were analyzed on a fresh weight (FW) basis while the clove powder, amla powder, kokum slice and kokum powder extracts were analyzed on a dry weight (DW) basis.

## Phenolic profile

The profile of the phenolic compounds was determined using the high-performance liquid chromatography (HPLC) assay method. The extracts were centrifuged at 13,500 rpm for 5 min after which 5 μL of the supernatant were injected using an Agilent ALS 1200 auto-extractor into an Agilent 1260 series (Agilent Technologies, Palo Alto, CA, USA) HPLC equipped with a D1100 CE diode array detector. The solvents used for gradient elution were 10 mM phosphoric acid (pH 2.5) and 100% methanol. The methanol concentration was increased to 60% for the first 8 min, then to 100% over the next 7 min, then decreased to 0% for the next 3 min and was maintained for 7 min with a total run time of 25 min per injected sample run. The analytical column used was Agilent Zorbax SB-C18, 250 – 4.6 mm i.d., with packing material of 5 μm particle size at a flow rate of 0.7 mL/min at room temperature. The absorbance values were recorded at 214 nm, 230 nm, 260 nm, and 306 nm and the chromatogram was integrated using Agilent Chem station enhanced integrator. Pure standards of benzoic acid, gallic acid, protocatechuic acid, ellagic acid, cinnamic acid, dihydroxybenzoic acid, p-coumaric acid, rutin, and catechin in 100% methanol were used to calibrate the respective standard curves

and retention times. The phenolic compounds detected in the extracts were expressed in microgram per gram fresh weight or dry weight (μg/g FW or DW). The amla slice, amla pickle, garlic slice and garlic pickle extracts were analyzed on a fresh weight (FW) basis while the clove powder, amla powder, kokum slice and kokum powder extracts were analyzed on a dry weight (DW) basis.

## Antimicrobial activity

The antimicrobial activity of the extracts was measured using the broth microdilution method based on the protocol as described by the Clinical and Laboratory Standards Institute [48]. A single colony of each bacterial strain was inoculated into 15 mL centrifuge tubes containing 10 mL Mueller-Hinton (MH) broth and the tubes were incubated overnight at 37°C. The overnight culture was centrifuged, and the bacterial pellet was resuspended in 10 mL phosphate-buffered saline (PBS) (VWR, Radnor, PA, USA). The turbidity of each culture was adjusted with PBS to a 0.5 McFarland standard with the help of a 0.5 McFarland standard reference solution (Remel, Thermo Fisher Scientific, Waltham, MA, USA) and the turbidity was confirmed using a UV-visible spectrophotometer (SmartSpec3000, Bio-Rad, Hercules, CA, USA). The adjusted cultures were then diluted in PBS at 1:20 dilution and used as the inoculum in the broth microdilution assay. A two-fold serial dilution of each filter-sterilized extract was done in microtiter plates using MH broth as the dilutant. For the control wells, sterile water was used instead of the extract. Around 10 μL of the adjusted bacterial inoculum were added to the respective wells and the microtiter plates were sealed with a plastic film and incubated for 16 h at 37°C in a microplate reader (Bio Tek Instruments, Agilent Technologies, USA). At every 15-min interval the microtiter plate was shaken for 5 seconds followed by a measurement of the absorbance of the wells at 600 nm. The absorbance values were plotted on a graph to get the growth curves and the minimal inhibitory concentration (MIC) of the extracts was expressed in mg GAE/g FW or DW. The amla slice, amla pickle, garlic slice and garlic pickle extracts were analyzed on a fresh weight (FW) basis while the clove powder, amla powder, kokum slice and kokum powder extracts were analyzed on a dry weight (DW) basis.

## Antioxidant activity

The antioxidant activity of the extracts was measured by their scavenging activity against the free radicals 2, 2-Dipheny-1-Picryl Hydrazyl (DPPH) (D9132-5G, Sigma-Aldrich, USA), and 2, 2-Azino-bis-(3-ethylbenzthiazoline-6-sulfonic acid) (ABTS) (A1888-5G, Sigma-Aldrich, USA) respectively. The DPPH scavenging assay was based on a protocol as described earlier [49] in which 0.25 mL of the extracts were added to 1.25 mL of 60 mM DPPH prepared in 95% ethanol while the controls had 0.25 mL of 95% ethanol instead of the extract. After 5 min of incubation, the extracts and their corresponding controls were centrifuged at 13,000 rpm for 1 min and the absorbance values of the supernatants were measured at 517 nm using a UV-visible spectrophotometer (Genesys 10S UV-VIS spectrophotometer, Thermo Scientific, NY, USA). The ABTS scavenging assay was based on a protocol as described earlier [50] in which 0.05 mL of the extracts was added to 1 mL of ABTS prepared in 95% ethanol while the controls had 0.05 mL of 95% ethanol instead of the extract. After 2 min of incubation, the extracts and their controls were centrifuged at 13,000 rpm for 1 min and the absorbance values of the supernatant were measured at 734 nm with a UV-visible spectrophotometer (Genesys 10S UV-VIS spectrophotometer, Thermo Scientific, NY, USA). The absorbance values from the DPPH and ABTS radical scavenging assays were used to calculate the percentage of

antioxidant activity for each extract using the following formula:

$$\% \text{ Antioxidant activity } = \frac{\text{Control}^{\text{absorbance}} - \text{Extract}^{\text{absorbance}}}{\text{Control}^{\text{absorbance}}} \times 100$$

Using a standard curve of different concentrations of Trolox (0.03 to 1mM) in 95% ethanol, the percentages of inhibitory activity obtained from the DPPH and ABTS radical scavenging assays were expressed as millimolar equivalents of trolox (mM TE).

## Alpha-amylase enzyme inhibitory activity

The α-amylase enzyme inhibitory activity of the extracts was measured based on a protocol as described earlier [49] and the activity was measured in a dose-dependent manner at undiluted, half, and one-fifth dilutions of the extracts. The extracts were diluted using distilled water and 500 μL of undiluted and diluted extracts were added to respective glass tubes, while the control tubes had 500 μL of 0.1M sodium phosphate buffer (containing 0.006M sodium chloride at pH 6.9) instead of the extract. Additionally, each extract had a corresponding blank tube containing 500 μL of the extract and 500 μL of the buffer instead of the enzyme. Then 500 μL of porcine pancreatic α-amylase (0.5 mg/mL buffer) (EC 3.2.1.1, purchased from Sigma Chemical Co, St Louis, MO, USA) was added to the extract and control tubes and incubated for 10 min at 25˚C. After incubation 500 μL of the substrate (1% starch in buffer) was added to all the tubes and incubated again for 10 min at 25˚C. Then 1 mL of 3, 5 dinitro salicylic acid was added and the tubes were incubated in a boiling water bath for 10 min to stop the reaction. After removing from the water bath and cooling down to room temperature, 10 mL of distilled water was added to all the tubes to ensure that the absorbance values in the control tubes ranged between 1.0 and 1.2, and the absorbance of all the tubes was measured at 540 nm using a UV-visible spectrophotometer (Genesys 10S UV-VIS spectrophotometer, Thermo Scientific, NY, USA). The absorbance values were then used to calculate the percentage of enzyme inhibitory activity of the extracts using the following formula:

$$\% \text{ Inhibition} = \frac{\text{Control}^{\text{absorbance}} - (\text{Extract}^{\text{absorbance}} - \text{Extract blank}^{\text{absorbance}})}{\text{Control}^{\text{absorbance}}} \times 100$$

Using a standard curve of different concentrations of acarbose (0.007 to 15 mM) in distilled water, the percentages of α-amylase inhibitory activity obtained from the enzyme inhibition assay were expressed as millimolar equivalents of acarbose (mM AE).

## Alpha-glucosidase enzyme inhibitory activity

The α-glucosidase enzyme inhibitory activity of the extracts was measured based on a protocol as described earlier [49] and the activity was measured in a dose-dependent manner at undiluted, half and one-fifth dilutions of the extracts. The extracts were diluted with 0.1M potassium phosphate buffer (pH 6.9) in 96 well microtiter plates in which 50 μL, 25 μL, and 10 μL of each extract were pipetted and the final volume made up to 50 μL by the addition of potassium phosphate buffer. Each extract had a corresponding control of 50 μL buffer instead of the sample and the volume in all the wells was made up to a final volume of 100 μL by the addition of 50 μL of the buffer. Then 100 μL of buffer containing yeast α-glucosidase enzyme (1 U/mL) (EC 3.2.1.20, purchased from Sigma Chemical Co, St. Louis, MO, USA) was added to each well and incubated for 10 min at 25˚C, after which 50 μL of the substrate, 5 mM p-nitrophenyl-α-D- glucopyranoside solution (prepared in buffer) was added to each well followed by 5-min incubation at 25˚C. The absorbance of all the wells was measured at 405 nm using a microplate

reader (Thermomax, Molecular device Co., VA, USA) at the 0- and 5-min time points of the 5-min incubation period, and the absorbance values were used to calculate the percentage of enzyme inhibitory activity using the following formula:

$$\% \text{ Inhibition} = \frac{(\text{Control}^{\text{abs}} \text{ 5 min} - \text{Control}^{\text{abs}} \text{ 0 min}) - (\text{Extract}^{\text{abs}} \text{ 5 min} - \text{Extract}^{\text{abs}} \text{ 0 min})}{(\text{Control}^{\text{abs}} \text{ 5 min} - \text{Control}^{\text{abs}} \text{ 0 min})} \text{ x } 100$$

Using a standard curve of different concentrations of Acarbose (0.007 to 15 mM) in distilled water, the percentages of α-glucosidase inhibitory activity obtained from the enzyme inhibition assay were expressed as millimolar equivalents of acarbose (mM AE).

## Angiotensin-I-converting enzyme (ACE) inhibitory activity

The ACE inhibitory activity of the extracts was measured based on a protocol as described earlier [49] and the activity was measured in a dose-dependent manner at undiluted, half, and one-fifth dilutions of the extracts. The extracts were diluted using distilled water. The substrate used was hippuryl-histidyl-leucine (HHL) and the enzyme ACE was obtained from rabbit lung (EC 3.4.15.1, purchased from Sigma Chemical Co, St Louis, MO, USA). To 50 μL of the extracts, 200 μL of 1M NaCl-borate buffer (pH 8.3) containing 2mU of ACE was added and then incubated at room temperature for 10 minutes. For the blank tubes, 50 μL of distilled water and 200 μL of buffer were used instead of the extract and enzyme. After this, 100 μL of 5mM HHL substrate (prepared in buffer) was added to all the tubes and incubated for one hour at 37˚C. The reaction was stopped by the addition of 150 μL of 0.5N HCl. The hippuric acid formed due to ACE activity was detected using the HPLC method for which, 5 μL of the reaction mixtures were injected using Agilent ALS 1200 autosampler into an Agilent 1260 series (Agilent Technologies, Palo Alto, CA, USA) HPLC equipped with a D1100 CE diode array detector. The solvents used for the gradient were a combination of 10mM phosphoric acid (pH2.5) and 100% methanol. For the first 8 min, the methanol concentration was increased to 60% then to 100% for 5 min, and finally to 0% for the next 5 min and the total run time was 18 min for each sample. The analytical column used was Agilent Zorbax SB-C18, 250 × 4.6 mm i.d., with packing material of 5 μm particle size at a flow rate of 0.7 mL/min at room temperature. The absorbance was measured at 228 nm and the chromatogram was integrated using Agilent Chemstation (Agilent Technologies) enhanced integrator for detection of hippuric acid. A hippuric acid standard was used to calibrate the standard curve and retention time and the percentage of ACE inhibition was calculated using the formula:

$$\% \text{ inhibition} = \frac{(\text{Control}^{\text{absorbance}} - \text{Extract}^{\text{absorbance}})}{(\text{Control}^{\text{absorbance}} - \text{Blank}^{\text{absorbance}})} \text{ x } 100$$

## Statistical analysis

The complete *in vitro* analysis was repeated four times. Analysis at every time point from each experiment was carried out in triplicates except for the antimicrobial assay which was done in duplicate. The mean, standard error, and standard deviation were calculated from replicates within the experiments using Microsoft Excel XP. The data was analyzed with analysis of variance (ANOVA) using Statistical Analytical Software (SAS version 9.4; SAS Institute, Cary, NC) and significant differences among extracts were determined by the Tukey's least mean square test at the 0.05 probability level.

## Results and discussion

### Total soluble phenolic content and phenolic profile

The TSP content of different forms of the selected botanicals ranged from 0.7 to 132.6 mg GAE/g FW or DW and significant differences in TSP content were observed among the extracts ($p < 0.05$) (Fig 1). The values of the TSP content of the botanical extracts are shown in S5 Table.

The TSP content of the amla slice, amla pickle, garlic slice and garlic pickle extracts were analyzed on a fresh weight (FW) basis while the clove powder, amla powder, kokum slice and kokum powder extracts were analyzed on a dry weight (DW) basis. The TSP content of the botanical extracts diluted with water at 1:40 dilution is shown in Fig 1. Amla powder had significantly higher TSP content at 132.6 mg GAE/g DW when compared to the rest of the extracts ($p < 0.05$) while garlic slice had the lowest TSP content at 0.7 mg GAE/g FW (Fig 1). The TSP content of amla can vary depending on the cultivar as well as the physical condition

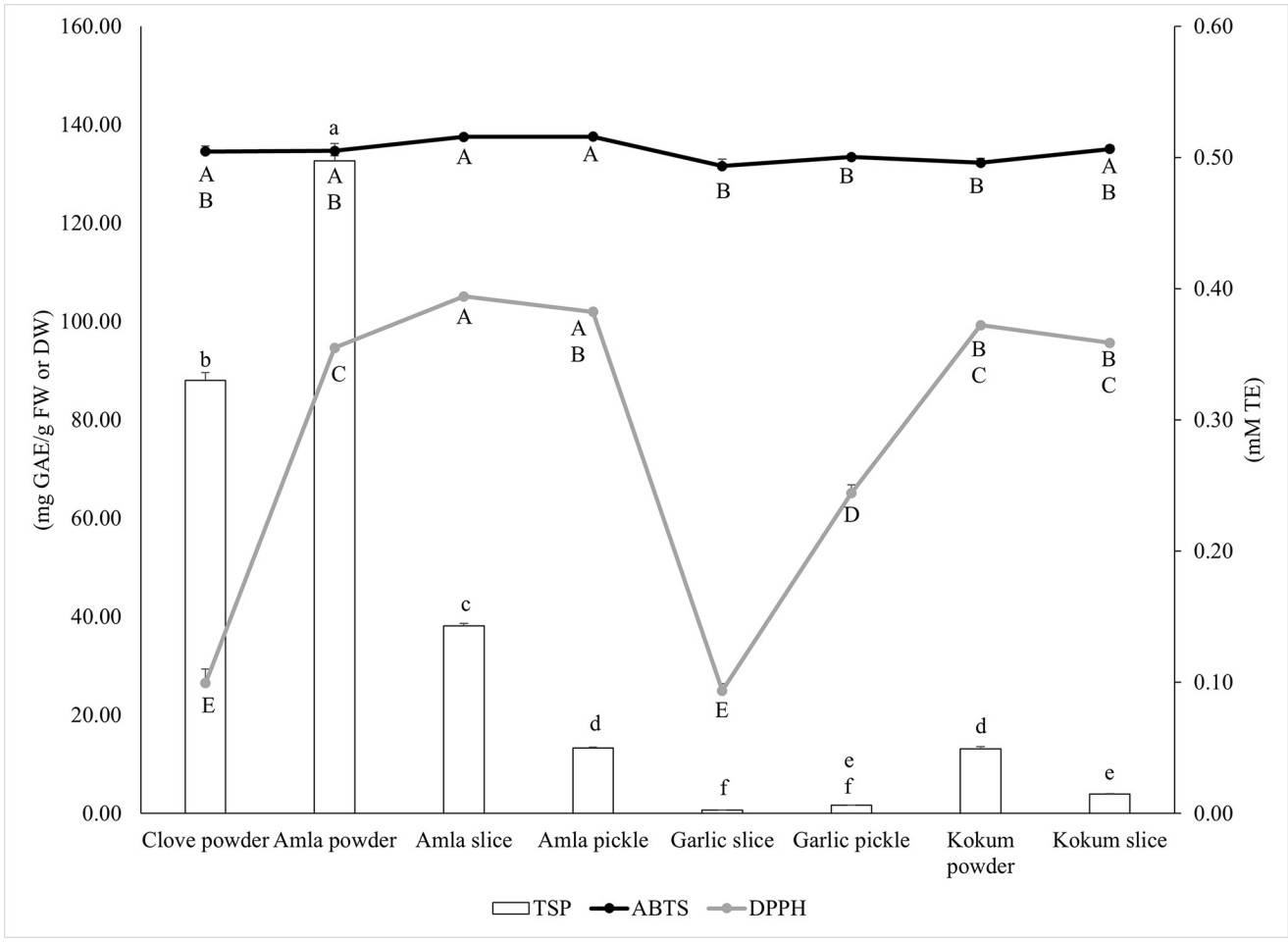

**Fig 1. Total soluble phenolic content of botanical extracts expressed in milligram gallic acid equivalents per gram fresh weight or dry weight (mg GAE/g FW or DW) and antioxidant activity of botanical extracts expressed in millimolar acarbose equivalents (mm AE).** Different lowercase letters indicate significant differences in TSP content among the extracts ($p < 0.05$). Different uppercase letters indicate significant differences in antioxidant activity (ABTS and DPPH- based) among the extracts ($p < 0.05$). The TSP content of the amla slice, amla pickle, garlic slice and garlic pickle extracts were analyzed on a fresh weight (FW) basis while the clove powder, amla powder, kokum slice and kokum powder extracts were analyzed on a dry weight (DW) basis.

**Table 1.** Phenolic profile of the botanical extracts expressed in microgram per gram fresh weight or dry weight (µg/g FW or DW).

| Extracts | Gallic acid[a,b,c] | Ellagic acid[a,b,c] | Cinnamic acid[a,b,c] | Dihydroxy benzoic acid[a,b,c] | Benzoic acid[a,b,c] | p-coumaric acid[a,b,c] | Protocatechuic acid[a,b,c] | Rutin[a,b,c] | Catechin[a,b,c] |
|---|---|---|---|---|---|---|---|---|---|
| Clove powder | 4.38 ± 0.6b | 10.16 ± 1.3c | 41.11 ± 1.2a | ND | ND | ND | ND | 2.24 ± 0.2 | 13.67 ± 1.4b |
| Amla powder | 4.51 ± 0.1b | 133.38 ± 6.6a | 5.94 ± 0.6b | ND | ND | ND | ND | ND | 1.88 ± 0.0d |
| Amla slice | 1.05 ± 0.1de | 30.93 ± 1.6b | 1.96 ± 0.1b | ND | 0.15 ± 0.0 | ND | ND | ND | 1.53 ± 0.1d |
| Amla pickle | 2.76 ± 0.2c | 6.27 ± 0.3c | 0.40 ± 0.0b | ND | ND | ND | ND | ND | 1.35 ± 0.0d |
| Garlic slice | 6.56 ± 0.1a | ND | ND | ND | ND | ND | ND | ND | 3.42 ± 0.1d |
| Garlic pickle | 1.67 ± 0.1d | ND | ND | 2.42 ± 0.2c | ND | 1.78 ± 0.1 | ND | ND | 2.93 ± 0.1d |
| Kokum powder | 0.83 ± 0.0de | ND | ND | 4.04 ± 0.0a | ND | ND | 2.44 ± 0.0a | ND | 24.01 ± 0.3a |
| Kokum slice | 0.54 ± 0.0e | ND | ND | 3.10 ± 0.0b | ND | ND | 1.29 ± 0.0b | ND | 10.18 ± 0.0c |

ND- Not Detected.

[a] Mean ± standard error.

[b] Different letters in each column indicate significant differences among extracts ($p<0.05$).

[c] The phenolic concentration of the amla slice, amla pickle, garlic slice and garlic pickle extracts were analyzed on a fresh weight (FW) basis while the clove powder, amla powder, kokum slice and kokum powder extracts were analyzed on a dry weight (DW) basis.

of the fruit (slices, juice, or powder) [51–53]. In the current study, although the TSP content of amla powder was significantly higher when compared to the other extracts ($p<0.05$), the content was still lower than what was reported in earlier studies [52,53]. The Folin-Ciocalteu (FC) reagent used in the estimation of TSP content can be affected by other compounds such as ascorbic acid, sugars and organic acids that may be present in plant-based food matrix [54]. Therefore, the high ascorbic acid content in amla can potentially interfere with the accurate estimation of TSP content. In the current study, serial dilution of the extracts has been done to avoid overestimation of TSP content of amla. The phenolic profile and content of the extracts varied greatly, and significant differences in the concentration of phenolic compounds detected were observed among the selected botanical extracts ($p<0.05$) (Table 1).

The phenolic profile of the amla slice, amla pickle, garlic slice and garlic pickle extracts were analyzed on a fresh weight (FW) basis while the clove powder, amla powder, kokum slice and kokum powder extracts were analyzed on a dry weight (DW) basis. The phenolic compounds detected were gallic acid (0.54 to 6.56 µg/g FW or DW), ellagic acid (6.27 to 133.38 µg/g FW or DW), cinnamic acid (0.40 to 41.11 µg/g FW or DW), dihydroxybenzoic acid (2.24 to 4.04 µg/g FW or DW), benzoic acid (0.15 µg/g FW or DW), p-coumaric acid (1.78 µg/g FW or DW), protocatechuic acid (1.29 to 2.44 µg/g FW or DW), rutin (2.24 µg/g FW or DW), and catechin (1.35 to 24.01 µg/g FW or DW). The garlic slice, amla powder, clove powder, and kokum powder extracts were found to have significantly higher concentrations of gallic acid, ellagic acid, cinnamic acid, and catechin, respectively ($p<0.05$) (Table 1). Benzoic acid, p-coumaric acid, and protocatechuic acid were detected only in the respective amla slice, garlic pickle, and kokum (powder and slice) extracts (Table 1). In an earlier study, jujube fruit was found to have a high concentration of gallic acid, 4-hydroxybenzoic acid, catechin and rutin, and the maturity stage of the fruit affected its phenolic content and associated bioactivity [1]. Overall, the TSP content and phenolic profile of the current study indicated that the selected

botanicals are good sources of phenolic phytochemicals, and the phenolic profile was found to vary among these botanicals. Furthermore, phytochemicals-rich fruits such as amla or kokum can be incorporated as fresh slices, powder, or pickles, in dietary strategies aimed at improving the health protective benefits.

## Antimicrobial activity

The antimicrobial activity of the botanical extracts was measured against serovars of *Salmonella*, *Listeria* and *E. coli* and the minimal inhibitory concentration (MIC) was expressed in mg GAE/g FW or DW. The antimicrobial activity of the amla slice, amla pickle, garlic slice and garlic pickle extracts were analyzed on a fresh weight (FW) basis while the clove powder, amla powder, kokum slice and kokum powder extracts were analyzed on a dry weight (DW) basis. The MIC of the extracts against *Salmonella*, *Listeria*, and *E. coli* serovars ranged from 0.25 to 22.00 mg GAE/g FW or DW, 0.12 to 11.00 mg GAE/g FW or DW, and 0.25 to 22.00 mg GAE/g FW or DW, respectively, and significant differences in MIC was observed among all the botanical extracts for all the microorganisms that were tested ($p<0.05$) (Table 2).

Among the *Salmonella* serovars (Enteritidis, Typhimurium, Montevideo, Stanley, and Saintpaul), extracts of kokum slice had lower MIC at 0.25 and 0.50 mg GAE/g DW ($p<0.05$), while extracts of clove had the highest MIC at 22.00 and 44.00 mg GAE/g DW (Table 2). No antimicrobial activity was detected for the garlic extracts (slice and pickle) against the *Salmonella* serovars Montevideo, Stanley, and Saintpaul. Among the *Listeria* serovars (1/2a, 1/2b, and 4b), extracts of kokum slice had lower MIC at 0.12 mg GAE/g DW ($p<0.05$), while

**Table 2. Minimal inhibitory concentration (MIC) of the botanical extracts expressed in milligram gallic acid equivalents per gram fresh weight or dry weight (mg GAE/g FW or DW).**

| Extracts | | | | | | Minimal inhibitory concentration[a,b,c] | | | | | |
|---|---|---|---|---|---|---|---|---|---|---|---|
| | *Salmonella* | | | | | *Listeria* | | | | *E. coli* | |
| | *S. enterica* subsp. *enterica* serovar Enteritidis | *S. enterica* subsp. *enterica* serovar Typhimurium | *S. enterica* subsp. *enterica* serovar Montevideo | *S. enterica* subsp. *enterica* serovar Stanley | *S. enterica* subsp. *enterica* serovar Saintpaul | *L. monocytogenes* serovar 1/2a | *L. monocytogenes* serovar 1/2b | *L. monocytogenes* serovar 1/2a | *L. monocytogenes* serovar 4b | *E. coli* serovar O157:H7 | *E. coli* serovar O26:H11 |
| Clove powder | 22.00 ± 0.0a | 22.00 ± 0.0a | 22.00 ± 0.0a | 44.00 ± 0.0a | 44.00 ± 0.0a | 11.00 ± 0.0a | NA | NA | NA | 22.00 ± 0.0a | 22.00 ± 0.0a |
| Amla powder | 8.29 ± 0.0c | 8.29 ± 0.0c | 8.29 ± 0.0b | 33.16 ± 0.0b | 16.58 ± 0.0c | 4.15 ± 0.0c | 4.15 ± 0.0b | 4.15 ± 0.0b | 4.15 ± 0.0b | 8.29 ± 0.0c | 8.29 ± 0.0c |
| Amla slice | 9.54 ± 0.0b | 9.54 ± 0.0b | 7.16 ± 2.4b | 19.09 ± 0.0c | 19.09 ± 0.0b | 4.77 ± 0.0b | 4.77 ± 0.0a | 4.77 ± 0.0a | 9.54 ± 0.0a | 9.54 ± 0.0b | 9.54 ± 0.0b |
| Amla pickle | 6.67 ± 0.0d | 6.67 ± 0.0d | 6.67 ± 0.0b | 6.67 ± 0.0d | 6.67 ± 0.0d | 3.33 ± 0.0d | 3.33 ± 0.0c | 3.33 ± 0.0c | 3.33 ± 0.0c | 6.67 ± 0.0d | 6.67 ± 0.0d |
| Garlic slice | 0.36 ± 0.0g | NA | NA | NA | NA | NA | NA | NA | NA | NA | NA |
| Garlic pickle | 0.85 ± 0.0f | 0.85 ± 0.0f | NA | NA | NA | NA | NA | NA | NA | 0.85 ± 0.0e | 0.85 ± 0.0e |
| Kokum powder | 1.65 ± 0.0e | 1.65 ± 0.0e | 3.30 ± 0.0bc | 3.30 ± 0.0e | 3.30 ± 0.0e | 0.82 ± 0.0e | 0.41 ± 0.0d | 0.41 ± 0.0d | 0.41 ± 0.0d | 0.82 ± 0.0f | 0.82 ± 0.0f |
| Kokum slice | 0.25 ± 0.0h | 0.25 ± 0.0g | 0.50 ± 0.0c | 0.50 ± 0.0f | 0.50 ± 0.0f | 0.12 ± 0.0f | 0.12 ± 0.0e | 0.12 ± 0.0e | 0.12 ± 0.0e | 0.25 ± 0.0g | 0.25 ± 0.0g |

NA- No Activity.

[a] Mean ± standard error.

[b] Different letters in each column indicate significant differences among the extracts ($p<0.05$).

[c] The antimicrobial activity of amla slice, amla pickle, garlic slice and garlic pickle extracts were analyzed on a fresh weight (FW) basis while the clove powder, amla powder, kokum slice, and kokum powder extracts were analyzed on a dry weight (DW) basis.

extracts of clove powder had the highest MIC at 11.00 mg GAE/g DW (Table 2). No antimicrobial activity was detected for the clove and garlic (slice and pickle) extracts against the *Listeria* serovars 1/2a, 1/2b, and 4b. Among the *E. coli* serovars (O157:H7 and O26:H11), extracts of kokum slice had lower MIC at 0.25 mg GAE/g DW ($p<0.05$), while extracts of clove powder had the highest MIC at 22.00 mg GAE/g DW (Table 2).

No antimicrobial activity was detected for the garlic slice and pickle extracts against most of the bacterial pathogens that were tested and overall, the *Listeria* serovars were more susceptible towards the antimicrobial activity of the selected botanicals as evident by the lower MIC values, when compared to *Salmonella* or *E. coli*. However, the garlic extracts (slice and pickle) were found to enhance the growth of the *Listeria* serovars. The growth curves of the four *Listeria* serovars in garlic extracts (slice and pickle) are shown in the supplemental figures (S1–S4 Figs) and the corresponding optical density values at 600mn are shown in the supplemental tables (S1–S4 Tables). Earlier studies have shown garlic extracts to display antimicrobial activity against different bacterial pathogens including *Listeria* [55,56], so hence in the current study, it was surprising to see garlic extracts enhancing the growth of *Listeria* instead of inhibiting it. Garlic can potentially show prebiotic activity that supports the growth of beneficial gut-related bacteria or probiotics [57], so it is possible that in the current study, the garlic extracts served as a prebiotic for the *Listeria* serovars that were tested. However, more investigation is needed to determine if this trend applies to only *Listeria* species and of the possible mechanisms behind this trend.

The mechanism of antimicrobial activity of phenolic phytochemicals is due to their ability to permeabilize the microbial cell membrane, inhibit DNA or protein synthesis, inhibit microbial metabolic activity, as well as chelate compounds required for microbial growth [58–61]. The antimicrobial activity observed in the current study clearly suggests that these selected botanicals, especially amla and kokum with their high baseline phenolic content, are good bioactive ingredients to counter bacterial contamination of other plant-based foods. This in the long term can potentially help mitigate the prevalence of foodborne illnesses.

## Antioxidant activity

The antioxidant activity of the extracts was measured via their ABTS and DPPH radical scavenging activity, and the activity was expressed in millimolar Trolox equivalents (mM TE). The ABTS and DPPH scavenging activity ranged from 0.49 to 0.52 mM TE/mL and 0.09 to 0.39 mM TE, respectively, and significant differences in radical scavenging activity were observed among the extracts ($p<0.05$) (Fig 1). The values of the ABTS and DPPH scavenging activity of the botanical extracts are shown in S6 and S7 Tables respectively. Amla slice and amla pickle extracts had higher ABTS scavenging activity at 0.52 mM TE ($p<0.05$), while garlic slice had the lowest ABTS scavenging at 0.49 mM TE. Similarly, amla slice extract had higher DPPH scavenging activity at 0.39 mM TE when compared to the rest of the botanical extracts ($p<0.05$), while garlic slice had the lowest DPPH scavenging activity at 0.09 mM TE (Fig 1). Overall, the selected botanicals had higher ABTS scavenging activity when compared to DPPH scavenging activity, which could be due to the difference in the chemical nature of these synthetic radicals. DPPH being more stable than ABTS, requires stronger antioxidant activity to quench the radical. In an earlier study, amla extracts showed high DPPH radical scavenging activity at 92.1% [62]. In another study, ice containing kokum was found to improve the oxidative stability and shelf life of chilled Indian mackerel (*Rastrelliger kanagurta*) [63]. These results indicate that amla and kokum fruit can be incorporated in dietary strategies as slices, powder, or pickle, due to their phytochemical-linked antioxidant activity which can offer chronic oxidative stress protective benefits for the management of common NCDs.

## Alpha-amylase and α-glucosidase enzyme inhibitory activity

The α-amylase and α-glucosidase inhibitory activity of the extracts was measured in a dose-dependent manner using undiluted, half-, and one-fifth dilutions of the extracts, and the inhibitory activity was expressed in millimolar acarbose equivalents (mM AE). The α-amylase inhibitory activity of the undiluted, half-, and one-fifth diluted extracts ranged from 0.03 to 0.19 mM AE, 0.00 to 0.19 mM AE, and 0.00 to 0.18 mM AE, respectively, and significant differences in α-amylase inhibitory activity were observed among the undiluted, half-, and one-fifth dilutions of the extracts ($p < 0.05$) (Table 3).

Among the undiluted extracts, the clove, kokum and amla extracts (slice and powder) had higher α-amylase inhibitory activity ranging from 0.18 to 0.19 mM AE ($p < 0.05$), while garlic slice had the lowest α-amylase inhibitory activity at 0.03 mM AE. The same trend was observed for the half-diluted and one-fifth-diluted extracts (Table 3). The α-glucosidase inhibitory activity of the undiluted, half-, and one-fifth diluted extracts ranged from 0.24 to 1.62 mM AE, 0.04 to 1.62 mM AE, and 0.00 to 1.61 mM AE, respectively, and significant differences in α-glucosidase inhibitory activity were observed among the undiluted, half-, and one-fifth-diluted extracts ($p < 0.05$) (Table 3). Among the undiluted extracts, the clove powder, kokum and amla extracts (slice and powder) had higher α-glucosidase inhibitory activity ranging from 1.59 to 1.62 mM AE ($p < 0.05$), while garlic slice had the lowest α-glucosidase inhibitory activity at 0.24 mM AE. A similar trend was observed for the half-and one-fifth dilutions of the extracts. Interestingly, extracts of amla and kokum (slices and powder) had the same level of α-amylase and α-glucosidase inhibitory activity, even at one-fifth dilutions of the extracts.

The amla and kokum (slice or powder) analyzed in the current study were found to display high α-amylase and α-glucosidase inhibitory activity even at lower dilutions of the extracts, thereby indicating their potent anti-hyperglycemic activity. In an earlier study, phenolic-rich fractions of kokum were found to have α-amylase inhibitory activity with $IC_{50}$ values ranging from 349.7 to 980.0 μg/mL [64]. In another study, extracts of amla at the highest concentration were found to have α-amylase and α-glucosidase inhibitory activity at 84.15% and 93.9% respectively [62]. Grape seed and tea extracts were found to have *in vitro* α-amylase and α-glucosidase

**Table 3. Alpha-amylase and α-glucosidase enzyme inhibitory activity of the botanical extracts expressed in millimolar acarbose equivalents (mM AE).**

| Extracts | α-amylase inhibitory activity[a,b] | | | α-glucosidase inhibitory activity[a,b] | | |
|---|---|---|---|---|---|---|
| | Undiluted | Half-diluted | One-fifth- diluted | Undiluted | Half-diluted | One-fifth- diluted |
| Clove powder | 0.18 ± 0.01ab | 0.17 ± 0.00ab | 0.06 ± 0.00c | 1.61 ± 0.01a | 1.61 ± 0.01a | 1.52 ± 0.02b |
| Amla powder | 0.19 ± 0.00a | 0.18 ± 0.01ab | 0.17 ± 0.00a | 1.62 ± 0.02a | 1.61 ± 0.00a | 1.59 ± 0.00a |
| Amla slice | 0.17 ± 0.01ab | 0.17 ± 0.00ab | 0.16 ± 0.00b | 1.62 ± 0.00a | 1.62 ± 0.00a | 1.54 ± 0.01b |
| Amla pickle | 0.15 ± 0.01c | 0.14 ± 0.01c | 0.03 ± 0.00d | 1.44 ± 0.03b | 1.22 ± 0.02b | 1.20 ± 0.01c |
| Garlic slice | 0.03 ± 0.00d | 0.005 ± 0.00d | NA | 0.24 ± 0.01d | 0.04 ± 0.03d | NA |
| Garlic pickle | 0.16 ± 0.01bc | 0.12 ± 0.01c | 0.03 ± 0.00d | 1.17 ± 0.01c | 0.80 ± 0.01c | 0.35 ± 0.01d |
| Kokum powder | 0.19 ± 0.00a | 0.19 ± 0.00a | 0.18 ± 0.00a | 1.61 ± 0.00a | 1.61 ± 0.01a | 1.61 ± 0.00a |
| Kokum slice | 0.19 ± 0.00a | 0.19 ± 0.00a | 0.18 ± 0.00a | 1.59 ± 0.01a | 1.59 ± 0.02a | 1.52 ± 0.01b |

NA- No Activity.

[a] Mean ± standard error.

[b] Different letters in each column indicate significant differences among extracts ($p < 0.05$).

inhibitory activity and the activity was attributed to the catechin content [65]. In the current study, catechin, a flavonoid compound with potential anti-hyperglycemic activity, was detected in all the botanical extracts. The results of the current *in vitro* assay model-based screening study indicate that amla and kokum (slice or powder) with high carbohydrate-digestive enzyme inhibitory potential can be incorporated into dietary strategies aimed at the management of chronic hyperglycemia, a common risk factor associated with type 2 diabetes. However, future clinical studies are required to further validate the anti-diabetic potential of amla and kokum and for its value-added integration in health targeted dietary and therapeutic applications.

### Angiotensin-I-converting enzyme (ACE) inhibitory activity

The ACE inhibitory activity was measured in a dose-dependent manner using undiluted, half-, and one-fifth dilutions of the extracts, and the inhibitory activity was expressed in percentages (%). The ACE inhibitory activity of the undiluted, half-diluted, and one-fifth diluted extracts ranged from 90.1 to 100%, 45.9 to 100%, and 0 to 100%, respectively, and significant differences in ACE inhibitory activity were observed among the undiluted, half-, and one-fifth-diluted extracts ($p < 0.05$) (Fig 2). The values of the ACE inhibitory activity of the botanical extracts are shown in S8 Table.

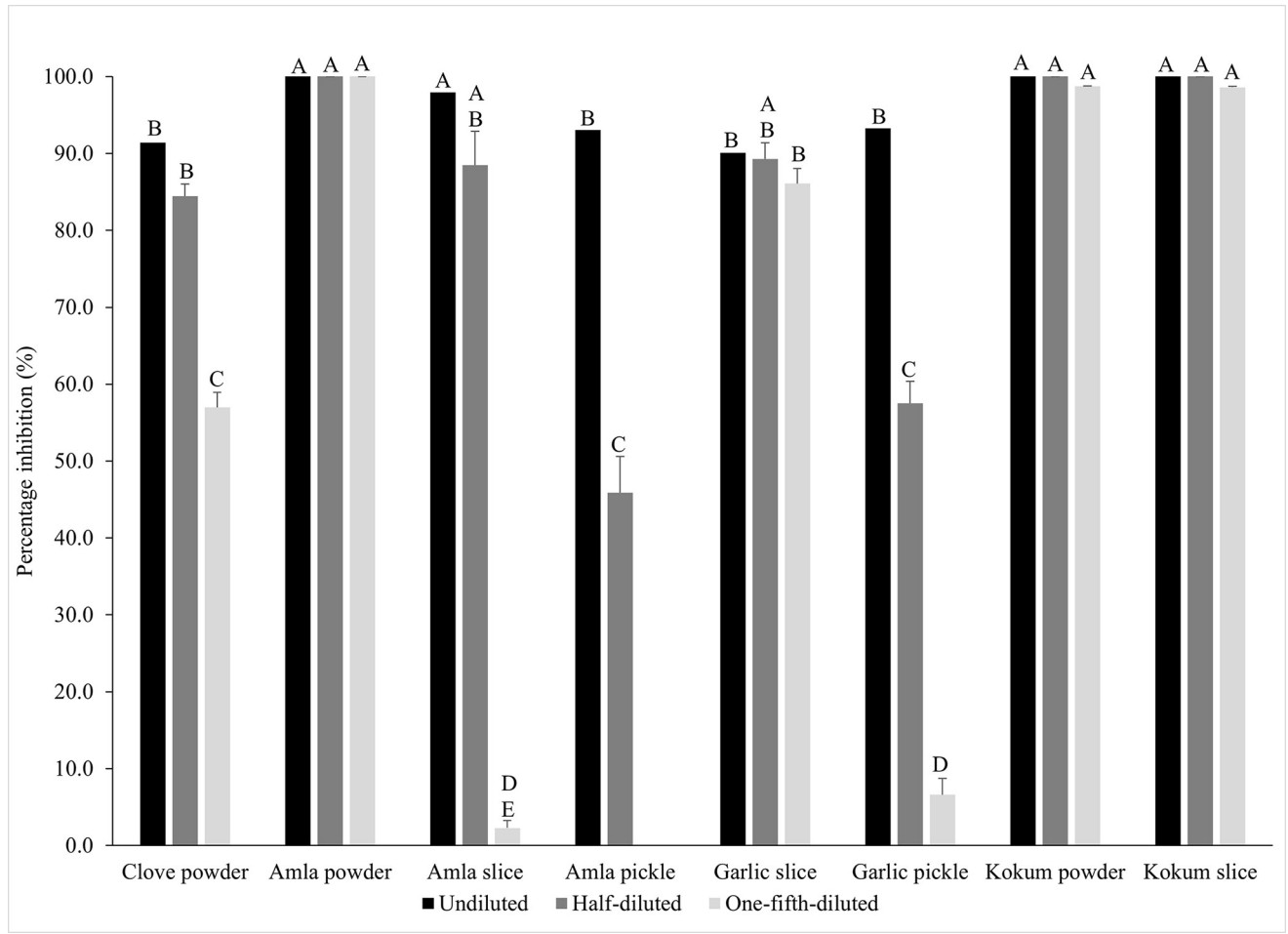

**Fig 2. Angiotensin-I-converting enzyme inhibitory activity of the botanical extracts expressed as percentages (%).** Different letters indicate significant differences among the extracts ($p < 0.05$).

Among the undiluted extracts, amla and kokum extracts (powder and slice) had significantly higher ACE inhibitory activity at 100% ($p<0.05$), while extracts of clove powder, amla pickle, and garlic (slice and pickle) had lower baseline ACE inhibitory activity ranging from 90.1 to 93.3%. (Fig 2). A similar trend was observed for the half-, and one-fifth diluted extracts. In the current study, the ACE inhibitory activity of the amla powder and kokum (powder and slice) extracts was found to be high even at one-fifth dilutions of the extracts, thereby indicating their potent antihypertensive activity. In a clinical study, an eight-week combination therapy of amla with antihypertensive drugs was found to significantly reduce the systolic blood pressure and diastolic blood pressure in patients with hypertension when compared to control (placebo) group [66]. In the current study, the *in vitro* ACE inhibition results indicates that amla and kokum, in addition to the other selected botanicals, can be incorporated in dietary support strategies as slices or powder to counter chronic hypertension commonly associated with type 2 diabetes and other NCDs.

## Conclusion

The screening of underutilized botanicals and plant-based foods including fruits, spices and other food ingredients is an important first step necessary for the selection and utilization of these phytochemicals-rich food-based botanicals with associated dual functional food safety-relevant and NCD health protective benefits. Under-utilized or indigenous fruits such as amla or kokum in different forms (slice, powder, or pickle) display NCD health protective properties via their antioxidant, antihyperglycemic, and antihypertensive activity, and hence can be potentially incorporated in dietary strategies to counter chronic inflammation, hyperglycemia, and hypertension. In addition to their health protective benefits, these phytochemicals-rich fruits or botanicals also have food safety-relevant benefits, specifically to counter foodborne illness related bacterial pathogens such as *Salmonella*, *Listeria*, and *E. coli*. However, there is always a possibility that certain food-based botanicals could potentially support the growth of these pathogens therefore requiring careful analysis of their antimicrobial properties. The formulation of different forms of amla and kokum (slice, powder, or pickle) with other bioactive plant-based foods can potentially reduce the risk of microbial contamination with these bacterial pathogens and help manage the burden of foodborne illnesses. Future studies can focus on formulation or dual functional synergy strategies involving amla, kokum, or clove with other bioactive plant-based foods in order to improve the phenolic phytochemicals-linked functional qualities for wider food safety and human health benefits.

## Supporting information

**S1 Fig. Growth curve of *L. monocytogenes* serovar 1/2a (10403S) in garlic slice and pickle extracts.**
(TIF)

**S2 Fig. Growth curve of *L. monocytogenes* serovar 1/2b (FSL J1-0194) in garlic slice and pickle extracts.**
(TIF)

**S3 Fig. Growth curve of *L. monocytogenes* serovar 1/2a (FSL F2-0515) in garlic slice and pickle extracts.**
(TIF)

**S4 Fig. Growth curve of *L. monocytogenes* serovar 4b (H7858) in garlic slice and pickle extracts.**
(TIF)

**S1 Table. Optical density values (OD 600nm) of *L. monocytogenes* serovar 1/2a (10403S) in garlic slice and pickle extracts.**
(DOCX)

**S2 Table. Optical density values (OD 600nm) of *L. monocytogenes* serovar 1/2b (FSL J1-0194) in garlic slice and pickle extracts.**
(DOCX)

**S3 Table. Optical density values (OD 600nm) of *L. monocytogenes* serovar 1/2a (FSL F2-0515) in garlic slice and pickle extracts.**
(DOCX)

**S4 Table. Optical density values (OD 600nm) of *L. monocytogenes* serovar 4b (H7858) in garlic slice and pickle extracts.**
(DOCX)

**S5 Table. Values of total soluble phenolic content of the selected botanical extracts.**
(DOCX)

**S6 Table. Values of ABTS- based antioxidant activity of the selected botanical extracts.**
(DOCX)

**S7 Table. Values of DPPH- based antioxidant activity of the selected botanical extracts.**
(DOCX)

**S8 Table. Values of angiotensin-I-converting enzyme inhibitory activity of the selected botanical extracts.**
(DOCX)

## Acknowledgments

The authors would like to thank Dr. Teresa Bergholz, Michigan State University, for her assistance in obtaining the bacterial strains. The authors would also like to thank Scott Hoselton, North Dakota State University, for his feedback regarding the antimicrobial assay.

## Author Contributions

**Conceptualization:** Ashish Christopher, Kalidas Shetty.

**Data curation:** Ashish Christopher.

**Formal analysis:** Ashish Christopher.

**Investigation:** Ashish Christopher, Kalidas Shetty.

**Methodology:** Ashish Christopher, Kalidas Shetty.

**Project administration:** Kalidas Shetty.

**Supervision:** Kalidas Shetty.

**Validation:** Ashish Christopher.

**Writing – original draft:** Ashish Christopher.

**Writing – review & editing:** Kalidas Shetty.

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
