## [Decision Letter · Decision Letter 0]

22 May 2024

PONE-D-24-12227Phytochemicals-Linked Dual Functional Food Safety and Human Health Protective Benefits of Select Food-Based BotanicalsPLOS ONE

Dear Dr. Shetty,

Thank you for submitting your manuscript to PLOS ONE. After careful consideration, we feel that it has merit but does not fully meet PLOS ONE’s publication criteria as it currently stands. Therefore, we invite you to submit a revised version of the manuscript that addresses the points raised during the review process.

In this manuscript, authors have studied about quantification of soluble phenolic content and phenolic profile in hot water extracts of amla (*Phyllanthus emblica*), clove (*Syzygium aromaticum*), kokum (*Garcinia indica*), and garlic (*Allium sativum*) using *in vitro *assay models as well as antimicrobial activity against strains of *Salmonella Enteritidis*, *Listeria monocytogenes*, and *Escherichia coli. *Authors have also studied about antioxidant, anti-hyperglycemic and anti hypertensive activity of the extracts was also determined using *in vitro *assay models and have reported that extracts have phenolic phytochemicals such as gallic, cinnamic, ellagic, benzoic, dihydroxybenzoic, protocatechuic, and *p*-coumaric acid along with catechin, rutin; significant antimicrobial activity against most of the bacterial strains; significant antioxidant,  anti-hyperglycemic and antihypertensive activity among the botanical extracts.

It is a good piece of research work however; the submitted manuscript in its current form is not acceptable for publication in the esteemed “Plos One” journal and requires minor revision. It is requested that authors must modify the manuscript according to reviewer’s comments. The corrections made in the manuscript should be highlighted. So, it would be easier to identify the modified content from the original submitted manuscript.

We look forward to receiving your revised manuscript.

Kind regards,

Pankaj Singh, Ph.D.

Academic Editor

PLOS ONE

A clean copy of the edited manuscript (uploaded as the new *manuscript* file).

Additional Editor Comments:

Editor’ Comments:

1.  Authors can add the name of chemicals used in the study in Chemicals used section.

2.  Authors have reported the values like rutin (2.24 μg/g FW or DW)……………, Authors must clearly mention in whole manuscript weather reported values are from fresh weight or dry weight sample.

3. In Table 2, the unit of MIC is missing.

4. Caption should be very clear regarding use of symbol. Authors must clearly explain the symbol used in the Figure and Table.

5. The significance levels in the Tables are not very much clear.

6. Please improve the image quality of Figure 2.

Reviewers' comments:

Reviewer's Responses to Questions

**Comments to the Author**

1. Is the manuscript technically sound, and do the data support the conclusions?

Reviewer #1: Yes

Reviewer #2: Yes

Reviewer #3: Yes

2. Has the statistical analysis been performed appropriately and rigorously? 

Reviewer #1: Yes

Reviewer #2: Yes

Reviewer #3: Yes

3. Have the authors made all data underlying the findings in their manuscript fully available?

Reviewer #1: Yes

Reviewer #2: Yes

Reviewer #3: Yes

4. Is the manuscript presented in an intelligible fashion and written in standard English?

Reviewer #1: Yes

Reviewer #2: Yes

Reviewer #3: Yes

5. Review Comments to the Author

Reviewer #1: Phytochemicals-Linked Dual Functional Food Safety and Human Health Protective

Benefits of Select Food-Based Botanicals

The following are my observations and recommendations for the improvement of the manuscript:

1) The Title of the manuscript needs a recast

2) Page 3, lines 46 - 51 of the introduction was a repetition of what was written in the abstract.

3) Page 3, line 51. NCDs can be listed

4) In the introduction section, a definition of phytochemicals-rich food-based botanicals can be provided for better understanding.

5) Page 3, line 62. Remove "another phytochemicals rich botanical"; Line 68: The word typhimurium is a serotype so must start with caps and not in italics. This should be corrected throughout the entire manuscript.

6) Page 4, line 81. Information about garlic should start in a new paragraph.

7) Heading 2.4: There was too much repetition of "used in this study" when highlighting the various bacterial strains

8) Replace "select" with "selected" in the whole manuscript.

9) line 290: This should be below Figure 1

10) line 308-309: I will suggest that this should start on a new paragraph for better clarity and organization of the manuscript

11) line 317-321: This is a repetition of information provided in Table 1. Discuss the results obtained in the Table with some existing literature and not repeat what is on the Table.

Page 31: line 290 to 296: The text should be pasted below Figure 2.

12) Page 21: Where this appeared: Instead, the garlic extracts enhanced the growth of these serovars when compared to the control (data not shown). I suggest that data on the garlic extract enhancing the growth of the Salmonella serovars should be briefly discussed to know how high the growth was during the incubation period. The mechanism behind this can also be checked and understood.

Overall, these are my recommendations for improvement of this manuscript:

1. The originality and significance of the article should be justified in line with existing literature to strengthen the arguments in the manuscript.

2. The clarity, organization, and coherence of the article should be improved

3. The writing style and structure of the article should be improved to allow the logical flow of ideas presented in the manuscript as suggested in (2) above.

Other concerns are marked out in the pdf format of the manuscript.

Reviewer #2: Dear Authors,

The research is scientifically relevant, however, small adjustments are necessary.

Considerations follow:

In the Materials and methods topic, it is essential to inform the concentration range of gallic acid used in the standard curve.

Reviewer #3: The manuscript is well-written and well-designed. The flow of the manuscript was very good. The abstract was very informative and brief. the introduction covered the required literatures related to the studied plants. The materials and methods are described in details. the data was well-presented and discussed very well- the conclusion is supporting the objective of the study. The tables and the figures are clear and presented the data very well. only very minor corrections in the attached manuscript need to be addressed.

6. PLOS authors have the option to publish the peer review history of their article (what does this mean?). If published, this will include your full peer review and any attached files.

Reviewer #1: **Yes: **Kolawole Banwo

Reviewer #2: No

Reviewer #3: No

---

## [Author Response · Author response to Decision Letter 0]

6 Jul 2024

Response to Reviewers, Editor and Journal Requirements

1. Please ensure that your manuscript meets PLOS ONE’s style requirements, including those for file naming.

Response: We have revised the manuscript according to the PLOS ONE’s style requirements, including those for file naming.

2. We suggest you thoroughly copyedit your manuscript for language usage, spelling, and grammar.

Response: We have revised the manuscript to improve language use, spelling and grammar.

Response: We have added a ‘Supporting information’ section and have included the values and standard error for TSP, ABTS, DPPH & ACE-1 inhibition assays that were used to make the figures.

4. We note that you have included the phrase “data not shown” in your manuscript.

Response: We have included the growth curves of the Listeria serovars in garlic extracts (slice and pickle) as supplemental figures. We have also included their corresponding optical density values (OD 600nm) as supplemental tables.

5. Please review your reference list to ensure that it is complete and correct.

Response: We have revised the ‘References’ section accordingly.

Additional Editor Comments:

Editor’ Comments and Response:

1. Authors can add the name of chemicals used in the study in Chemicals used section.

Response: We have added the names of the chemicals, media and enzymes in the ‘Chemical used’ section.

2. Authors have reported the values like rutin (2.24 μg/g FW or DW)……………, Authors must clearly mention in whole manuscript weather reported values are from fresh weight or dry weight sample.

Response: We have included the statement ‘The amla slice, amla pickle, garlic slice and garlic pickle extracts were analyzed on a fresh weight (FW) basis while the clove, amla powder, kokum slice and kokum powder extracts were analyzed on a dry weight (DW) basis.’ This statement was included in the ‘Materials & Method’ and ‘Result & Discussion’ sections for the experiments done on total soluble phenolic content, phenolic profile & antimicrobial activity as these results were expressed in fresh or dry weight basis. 

3. In Table 2, the unit of MIC is missing.

Response: We have verified that the unit of MIC is mentioned in the Table 2 caption. The caption is as follows: ‘Minimal inhibitory concentration (MIC) of the botanical extracts expressed in milligram gallic acid equivalents per gram fresh weight or dry weight (mg GAE/g FW or DW).’

4. Caption should be very clear regarding use of symbol. Authors must clearly explain the symbol used in the Figure and Table.

Response: The caption for Figure-1 has been revised as follows: ‘Total soluble phenolic content of botanical extracts expressed in milligram gallic acid equivalents per gram fresh weight or dry weight (mg GAE/g FW or DW) and antioxidant activity of botanical extracts expressed in millimolar acarbose equivalents (mm AE)’. We have verified that the symbols used in the Figures and Tables are explained either in the Figure caption or in the Table footnote.

5. The significance levels in the Tables are not very clear.

Response: We have mentioned the statement: ‘Different letters in each column indicate significant differences among extracts (p<0.05).’ This statement is mentioned in the footnote of Table-1 & 2 and in the caption of Figure-2. In the case of Figure-1 caption, we have mentioned the statement ‘Different lowercase letters indicate significant differences in TSP content among the extracts (p<0.05). Different uppercase letters indicate significant differences in antioxidant activity (ABTS and DPPH- based) among the extracts (p<0.05).’ This is to help the reader distinguish between the significant levels for the TSP content & antioxidant activity (DPPH and ABTS-based) that are present within the same graph. 

6. Please improve the image quality of Figure 2.

Response: The quality of all the figures (Fig-1, Fig-2 and supplemental figures) have been revised according to guidelines.

Reviewer -1 Comments and Response:

1. The Title of the manuscript needs a recast.

Response: The title has been revised to ‘Phytochemical-linked Food Safety and Human Health Protective Benefits of the Selected Food-Based Botanicals.’

2. Page 3, lines 46 - 51 of the introduction was a repetition of what was written in the abstract.

Response: The introduction has been revised to avoid repetition with the abstract. 

3. Page 3, line 51. NCDs can be listed

Response: The NCDs that are the focus of this manuscript have been listed. The list includes type 2 diabetes, hypertension and the associated oxidative stress.

4. In the introduction section, a definition of phytochemicals-rich food-based botanicals can be provided for better understanding.

Response: The ‘Introduction’ section has been revised to give the reader better understanding on phytochemical-rich foods.

5. Page 3, line 62. Remove "another phytochemicals rich botanical"; Line 68: The word typhimurium is a serotype so must start with caps and not in italics. This should be corrected throughout the entire manuscript. 

Response: The line ‘another phytochemical rich botanical’ has been removed from page 3, line 62. We have verified that the word Typhimurium begins with a capital letter throughout the manuscript.

6. Page 4, line 81. Information about garlic should start in a new paragraph.

Response: The information on garlic has been started in a new paragraph.

7. Heading 2.4: There was too much repetition of "used in this study" when highlighting the various bacterial strains

Response: We have used other statements such as ‘analyzed in this study’ and ‘tested in this study’.

8. Replace "select" with "selected" in the whole manuscript.

Response: We have replaced ‘select’ with ‘selected’ in the whole manuscript.

9. line 290: This should be below Figure 1

Response: After careful review we have decided not to make any changes in order to preserve the logical flow of the statement.

10. line 308-309: I will suggest that this should start on a new paragraph for better clarity and organization of the manuscript

Response: After careful review we have decided not to make any changes in order to preserve the logical flow of the statement.

11. line 317-321: This is a repetition of information provided in Table 1. Discuss the results obtained in the Table with some existing literature and not repeat what is on the Table. Page 31: line 290 to 296: The text should be pasted below Figure 2.

Response: We have revised the information in ‘Total soluble phenolic content and phenolic profile’ section to include a reference on phenolic profile of another study. 

12. Page 21: Where this appeared: Instead, the garlic extracts enhanced the growth of these serovars when compared to the control (data not shown). I suggest that data on the garlic extract enhancing the growth of the Salmonella serovars should be briefly discussed to know how high the growth was during the incubation period. The mechanism behind this can also be checked and understood.

Response: After re-examining the data we found that the garlic extracts (slices and pickle) enhanced the growth of the Listeria serovars and not the Salmonella serovars. We have revised the section accordingly and have included the growth curves of the Listeria serovars in garlic extracts (slice and pickle) as supplemental figures. We have also included the corresponding optical density values (OD 600nm) as supplemental tables.

Reviewer -2 Comments and Response:

1. Dear Authors, the research is scientifically relevant, however, small adjustments are necessary. Considerations follow: In the Materials and methods topic, it is essential to inform the concentration range of gallic acid used in the standard curve.

Response: We have mentioned the concentration range of gallic acid, trolox and acarbose used for the preparation of the standard curves in the ‘Material & methods’ section.

Reviewer-3 comments:

1. The manuscript is well-written and well-designed. The flow of the manuscript was very good. The abstract was very informative and brief. the introduction covered the required literatures related to the studied plants. The materials and methods are described in details. the data was well-presented and discussed very well- the conclusion is supporting the objective of the study. The tables and the figures are clear and presented the data very well. only very minor corrections in the attached manuscript need to be addressed.

Response: We would like to thank Reviewer-3 for the feedback. We have made the necessary minor corrections based on comments from the Editor and the other reviewers.

Additional Authors’ comments to the editor and further improvements and clarifications to the manuscript:

1. The following statement was added to the ‘Antioxidant Activity’ in the ‘Materials & Methods’ section: ‘Using a standard curve of different concentrations of Trolox in 95 % ethanol, the percentages of inhibitory activity obtained from the DPPH and ABTS radical scavenging assays were expressed as millimolar equivalents of trolox (mM TE).’

2. The following statement was added to ‘Total soluble phenolic content and phenolic profile’ in the ‘Results & Discussion’ section: ‘The TSP content of botanical extracts diluted with water at 1:40 dilution is shown in Fig 1.’

3. The phrase ‘statistical differences’ has been replaced with ‘significant differences’ in the Results & discussion’ section.

4. We have added two of the references suggested by the editor/ reviewer in the ‘Introduction’ and ‘Results & Discussion’ sections of the manuscript. 

5. We have added the following statement to the ‘Conclusion’ section: ‘However, there is always a possibility that certain food-based botanicals could potentially support the growth of these pathogens therefore requiring careful analysis of their antimicrobial properties’. 

6. We have added a ‘Supporting information’ section which has captions of supplemental tables and figures.

7. We have also included in-text citation of the supplemental tables and figures.

8. A minor correction was made to citation 8 in ‘References’ section.

9. A minor correction was made to citation 10 in ‘References’ section.

10. A minor correction was made to citation 14 in ‘References’ section.

11. Two citations (Saeed & Tariq 2007; Singh et al 2019) have been removed from the ‘References’ section.

12. The URL link for citation 25 was removed in the ‘References’ section.

13. A minor correction was made to citation 30 in ‘References’ section.

14. A minor correction was made to citation 34 in ‘References’ section.

15. A minor correction was made to citation 36 in ‘References’ section.

16. A minor correction was made to citation 37 in ‘References’ section.

17. A minor correction was made to citation 38 in ‘References’ section.

18. A minor correction was made to citation 41 in ‘References’ section.

19. A minor correction was made to citation 45 in ‘References’ section.

20. A minor correction was made to citation 56 in ‘References’ section.

21. A minor correction was made to citation 60 in ‘References’ section.

22. The numbers for the level 1 and level 2 headings were removed.

---

## [Editor Report · Decision Letter 1]

12 Jul 2024

Phytochemical-Linked Food Safety and Human Health Protective Benefits of the Selected Food-Based Botanicals

PONE-D-24-12227R1

Dear Dr. Shetty,

We’re pleased to inform you that your manuscript has been judged scientifically suitable for publication and will be formally accepted for publication once it meets all outstanding technical requirements.

Kind regards,

Pankaj Singh, Ph.D.

Academic Editor

PLOS ONE
---

## [Editor Report · Acceptance letter]

19 Jul 2024

PONE-D-24-12227R1 

PLOS ONE

Dear Dr. Shetty, 

I'm pleased to inform you that your manuscript has been deemed suitable for publication in PLOS ONE. Congratulations! Your manuscript is now being handed over to our production team.

Kind regards, 

on behalf of

Dr. Pankaj Singh 

Academic Editor

PLOS ONE